

**Title**
Reviews and Syntheses: Impacts of plant silica - herbivore interactions on terrestrial biogeochemical
cycling
**Authors**
Bernice C. Hwang[1] and Daniel B. Metcalfe[1,2]
**Affiliations**
[1]Lund University, Department of Physical Geography and Ecosystem Science, Sölvegaten 12, Lund, 223 62,
Sweden.
[2]Umeå University, Department of Ecology and Environmental Science, Linnaeus väg 6, Umeå, 901 87,
Sweden.
**Correspondence**
Bernice C. Hwang
E-mail: bernice.hwang@nateko.lu.se
**Funding Information**
European Research Council (ECOHERB, grant no. 682707)
**Keywords**
Carbon, herbivory, nutrient cycling, plant defense, plant-soil processes, silica, stoichiometry



**Abstract**
Researchers have known for decades that silicon plays a major role in biogeochemical and plant-soil
processes in terrestrial systems. Meanwhile, plant biologists continue to uncover a growing list of benefits
derived from silicon to combat abiotic and biotic stresses, such as defense against herbivory. Yet despite
growing recognition of herbivores as important ecosystem engineers, many major gaps remain in our
understanding of how silicon and herbivory interact to shape biogeochemical processes, particularly in
natural systems. We review and synthesize 119 available studies directly investigating silicon and herbivory
to summarize key trends and highlight research gaps and opportunities. Categorizing studies by multiple
ecosystem, plant, and herbivore characteristics, we find substantial evidence for a wide variety of
important interactions between plant silicon and herbivory, but highlight the need for more research
particularly in non-graminoid dominated vegetation outside of the temperate biome as well as on the
potential effects of herbivory on silicon cycling. Continuing to overlook silicon-herbivory dynamics in
natural ecosystems limits our understanding of potentially critical animal-plant-soil feedbacks necessary
to inform land management decisions and to refine global models of environmental change.



## 1 Introduction

For centuries researchers have intensively studied cycles of key nutrients influencing plant growth and diversity such as nitrogen (N) and phosphorus (P) across a diverse range of ecosystems (*e.g.*, Elser et al., 2007). Meanwhile, studies have often overlooked other nutrients such as silicon (Si), which is important for plant function and protection (Cooke et al., 2016) as well as for biogeochemical cycling (Street-Parrott and Barker, 2008). For instance, Si can reduce the impact of many abiotic and biotic pressures, including water, temperature and salinity stress, as well as nutrient deficiency, heavy metal toxicity, disease, and herbivory (Debona et al., 2017). Plants can likewise affect terrestrial silica fluxes by controlling weathering rates and activity of dissolved Si in soils and streams (Derry et al., 2005). Plants and their associated microbiota can accelerate the weathering of silicate minerals by altering the physical properties and reactivity of the soil environment (*e.g.*, pH, moisture, exposed surface area of minerals), and by taking up essential nutrients which destabilizes silicate minerals (Drever, 1994; Street-Parrott and Barker, 2008 and references therein). Since the silicate weathering process consumes carbon dioxide ($CO_2$) through chemical weathering of calcium- and magnesium-silicate minerals in continental rocks, the effects of plants on the Si cycle may influence the global carbon (C) cycle (Street-Perrott and Barker, 2008).

Plants take up dissolved Si from direct weathering of mineral silicates and biogenic Si released from leaf litter to soil (Cornelis and Delvaux, 2016), and Si in plants exceeds the concentrations of many plant macro-nutrients (Epstein, 1999). Si accumulation varies among species, with some plants containing only trace amounts, whereas Si can constitute up to 10% plant dry mass in "high Si accumulators" such as many grasses (Hodson et al., 2005). In the last decade, many studies have focused on agricultural systems (Guntzer et al., 2012; see also Fig. 1), in particular looking at grass species (*e.g.*, McNaughton et al., 1985; Hartley and DeGabriel, 2016), with fewer studies examining the role or importance of plant Si in natural (*i.e.*, non-agricultural and in the field) environments even though the degree of dissolved Si passed through ecosystems as filters can vary dramatically by biome (Cooke and Leishman, 2011; Schoelynck et al., 2014;





Cornelis et al., 2016; see also Fig 1). In natural systems, studies have focused on graminoid-dominated
vegetation types in the temperate biome, with relatively little research in tropical and boreal/subarctic
forest and herbaceous vegetation types, even though plant Si uptake and storage in these systems may be
significant (Cornelis et al., 2010; Katz, 2014).
Herbivory also creates a number of important feedbacks between plants and soils (Bardgett and Wardle,
2003). For example, invertebrate herbivores can supply a remarkably large amount of nutrients to tropical
and subarctic systems compared to other major sources, bringing likely labile nutrients to the forest floor
in the form of frass and bodies (Hartley and Jones, 2004; Metcalfe et al., 2013; Kristensen et al., 2018).
Selective herbivory may also result in the dominance of plants that are nutrient poor and/or better
defended, which ultimately produces more recalcitrant litter that decomposes more slowly (Bardgett and
Wardle, 2003). Since plant Si is involved in plant defense (Debona et al., 2017), interactions between
herbivory and plant Si (Quigley and Anderson, 2016) may have the potential to exert a powerful influence
over ecosystem biogeochemistry and function.
There are multiple detailed reviews covering silicon terrestrial biogeochemistry (*e.g.*, Conley, 2002; Struyf
and Conley, 2012) as well as Si-derived benefits to plants such as herbivore defense (*e.g.*, Frew et al., 2018;
Katz, 2019). However, given the likely importance of Si in multiple terrestrial biomes and the strong
evidence for a tight coupling between plant Si and herbivory, remarkably few studies have investigated Si-
herbivory dynamics in the context of biogeochemical cycling. Here, we review the role of plant Si-herbivore
dynamics in biogeochemical cycles in order to summarize existing knowledge and emerging patterns,
identify gaps in knowledge, and describe future research priorities. Towards this effort, we surveyed
available literature between 1900-2020 in the Web of Science core collection database using search terms
"silic*" and "herbivor*" and not "in silico" (314 results). We then filtered the results until only those
publications that directly studied Si and herbivory remained (119 publications), which we categorized into
various ecosystem, plant, and herbivore characteristics (Fig. 1). Our purpose is to identify areas where this



knowledge can be currently useful, such as agricultural and land management, and assist efforts to better
integrate potentially important but overlooked herbivore-plant-soil interactions into global
biogeochemical models to more accurately predict ecosystem function shifts in the face of environmental
change (Van der Putten et al., 2013).
**2 Silicon in terrestrial systems**
The biogeochemical Si cycle impacts global $CO_2$ concentrations through weathering silicate minerals and
transferring $CO_2$ from the atmosphere to the lithosphere (Conley 2002). In terrestrial systems, soil is the
primary source of plant Si, with global variability of soil-forming factors (*e.g.*, parent rock, climate,
topography, age, biota) explaining the large variability in Si cycling rates (Cornelis et al., 2016). Many plant
species accumulate Si predominantly in leaves and needles as amorphous, biogenic silica, in large discrete
bodies known as phytoliths. Si then returns to soil when plant material decomposes either as dissolved Si,
a quickly-available source of Si for terrestrial plants, or as phytoliths, where C incorporated by phytoliths
may accumulate in soils and sediments for hundreds to thousands of years. For this reason, phytolith
accumulation is considered as a key mechanism of biogeochemical C sequestration (Parr and Sullivan,
2005). Plant-accumulated Si has been shown to reduce the magnitude of Si released from terrestrial to
aquatic ecosystems, thereby having direct implications on Si availability in coastal waters, which could the
influence diatom blooms and C uptake rates (Coney et al., 2008, Carey and Fulweiler, 2012). Ultimately,
terrestrial systems supply approximately 78% of annual silica inputs to oceans (Tréguer and De La Rocha,
2013) and Si-accumulating vegetation accounts for 55% of terrestrial net primary productivity (33 GtC per
year), on par with the rate that marine diatoms sequester C (Conley, 2002; Carey and Fulweiler, 2012).
Plants may preferentially use Si for certain functions such as structure and defense instead of C (Cooke
and Leishman, 2012) as Si can provide plants with structural support at a lower metabolic cost than C
(Raven, 1983). In some cases, Si is negatively correlated with lignin and cellulose content, possibly
contributing to plant structural support, and phenol content (Schoelynck et al., 2010; Cooke and Leishman,



2012). Intermediate Si fertilization additions, for example, have enhanced aboveground growth for crops
and reeds, possibly due to a partial substitution of organic C compounds by Si in plant tissue (Schaller et
al., 2012; Neu et al., 2017).
Soil Si may also facilitate the acquisition and release of other essential plant nutrients. For example, a
laboratory experiment demonstrated that silicon addition significantly increased P mobilization in a variety
of arctic soils (Schaller et al., 2019). Researchers have found that Si can also increase plant N use efficiency
while decreasing C and increasing P in grasslands (Neu et al., 2017) and affect the calcium (Ca) content of
grasses (Brackhage et al., 2013). The abundance of certain plant functional groups (*e.g.*, Si-rich grasses) in
plant communities can affect Si and Ca biogeochemistry as a result of differences in elemental
concentrations among plant species and related effects on nutrient cycling via processes such as
decomposition (Schaller et al., 2016). The potential of Si to affect plant growth and elemental
stoichiometry in grasslands can, by extension, then affect biogeochemical cycles (Schaller et al., 2017).
Changes in plant nutrient stoichiometry due to Si may have broad implications for other natural systems,
where the (un)availability of essential nutrients can shape the productivity, composition, diversity,
dynamics and interactions of plant, animal, and microbial populations (Vitousek, 2010). For example, P can
be limiting in weathered tropical forest soils (Vitousek, 2010), so changes in P availability due to Si (Neu et
al., 2017) can have concomitant effects on productivity. Furthermore, the recycling of Si within forests
impacts continental Si cycling, especially in tropical forests which take up Si at a faster rate in terms of
mass per unit ground area than other biomes, particularly in highly weathered soils (Alexandre et al., 1997;
Cornelis et al., 2016; Schaller et al., 2018). Notably, unlike major plant nutrients for which foliar
concentrations significantly decreased with increasing soil age, foliar Si concentrations continually
increased with increasing soil age in two Australian soil chronosequences (de Tombeur et al., 2020).
Returning phytoliths to topsoil can result in the slow-release of Si that sustains the terrestrial cycle during
ecosystem retrogression.



A summary of literature shows that studies of Si-herbivore dynamics have focused on the ecology and
physiology of Si in the grass family (Fig. 1), *Poaceae*, which includes many species that accumulate large
amounts of Si in their tissues. Species richness can increase plant Si stocks via its positive relationship with
biomass production but can have a negative effect on Si concentration in the aboveground biomass, which
may influence processes such as decomposition, nutrient cycling and herbivory (Schaller et al., 2016).
Some studies indicate Si content of plant litter may be positively correlated to decomposition rate (*e.g.*,
Schaller et al., 2013). Because plant-stored Si releases Si into soils and sediments relatively rapidly, high Si-
accumulating plants like grasses can influence Si turnover rates in ecosystems by uptake, storage, and
release of Si during plant decomposition (Schaller et al., 2016 and references therein).
However, several non-monocot angiosperms also store significant foliar Si (Hodson et al., 2005), and
variation in foliar Si can play important roles such as in plant defense and metal toxicity reduction even for
low-accumulators (Katz, 2014 and references therein). Si contents vary by as much as 2-3 orders of
magnitude among plant families, orders and phyla (Hodson et al., 2005), and grass-rich systems tend to
be richer in Si and more productive than systems without grasses (Carey and Fulweiler, 2012). However,
nutrient use strategies can vary intra-specifically across environmental gradients at least within controlled
settings (Harley and DeGabriel, 2016 and references therein). Therefore, we need more field-based
information about how Si content varies along large-scale environmental gradients to improve global
biogeochemistry models.
Anthropogenic perturbations, such as agriculture, deforestation, urbanization, and climate warming, can
also have profound effects on terrestrial silica biogeochemistry (Conley et al., 2008; Struyf and Conley,
2012; Carey and Fulweiler, 2016; Gewirtzman et al., 2019). Deforested areas can increase soil erosion,
resulting in the loss of high biogenic Si concentrations found in surface soils (Saccone et al., 2007). Urban
areas have limited ability to take up dissolved Si into vegetation and agricultural lands retain less biogenic
Si as it is frequently removed through harvesting and may not be replenished by vegetation-stimulated



silicate weathering (Struyf and Conley, 2012; Vandevenne et al., 2012). Global agricultural Si export from
harvesting is estimated to be 223 kg Si yr$^{-1}$ (Matichenkov and Bocharnikova, 2001) compared to the 142
kg Si yr$^{-1}$ total quantity of dissolved Si transferred from continents to oceans by rivers (Tréguer et al., 1995).
In addition, soil warming due to climate change can increase the extent of internal Si recycling in temperate
forests (Gewirtzman et al., 2019) and changes in precipitation intensity as expected with climate change
can increase surface run-off and top-soil erosion reducing biogenic Si in surface soils (Conley et al., 2008;
Struyf et al., 2010). Thus, researchers should take into account the potential impact of land-use and climate
changes on terrestrial Si fluxes when modeling terrestrial Si mobilization.
**3 Effects of silicon on herbivory**
Si is known to defend plants against a wide range of biotic stresses, including pathogen infection and
herbivory (Reynolds et al., 2009; Guntzer et al., 2012; Frew et al., 2018 and references therein). Si-
mediated defenses against herbivores involve both direct and indirect physical or mechanical barriers, as
well as indirect biochemical or molecular mechanisms (Fig. 2). In addition, plant communities with high Si
can affect herbivore communities. Plant groups with high leaf toughness, high Si concentrations, and low
leaf nitrogen concentrations, for example, can also be associated with decreased grassland herbivore
species richness (Schuldt et al., 2019). Furthermore, studies have reported how grass Si content can drive
herbivore populations, which may be synchronized with plant Si content cycles (Massey et al., 2008;
Hartley, 2015).
Si-derived mechanical barriers are often thought to effectively shorten the duration of attack, both directly
or indirectly, by making tissues more difficult to chew, penetrate, and digest (Hunt et al., 2008; Massey
and Hartley, 2009), and increasing exposure time to predators (Massey and Hartley 2009). For example,
discrete silica bodies in and on the surface of leaves can reduce herbivory (Hartley et al., 2015). Si
structures may wear down herbivore mouthparts (Massey and Hartley 2009), affecting herbivore ingestion
and nutrition (Frew et al., 2018; Hunt et al., 2008). These abrasive phytoliths can also lacerate herbivore





body parts or facilitate pathogen transmission into herbivores (Lev-Yadun and Halpern 2019). Si may also
alter nutritional quality indirectly via changes to foliar C : N ratio and P concentrations (Frew et al., 2018)
while plant nutrient status may influence the overall efficacy of Si-based defenses against herbivory (Mosie
et al., 2019). As a consequence, some herbivores demonstrate low preference for Si-rich plants and slower
growth rates when feeding on Si-rich diets (Massey and Hartley, 2006, 2009).
Several studies have documented effects of soil Si addition on plant chemical defenses (*e.g.*, Reynolds et
al., 2009), anti-herbivore phytohormonal signaling (*e.g.*, Hall et al., 2019), and changes in plant nutritional
quality (*e.g.*, Frew et al., 2018; Moise et al., 2019). For example, Si may induce indirect defense
mechanisms by altering the composition of herbivore-induced plant volatiles that attract herbivore
parasitoids and predators (Liu et al., 2017). Some studies also point to effects of Si on plant secondary
metabolism and gene expression in plant development and defense (Markovich et al., 2017; Frew et al.,
2018 and references therein). But because Si has relatively limited chemical reactivity, its role on plant
chemical changes may be indirect rather than direct (Coskun et al., 2019).  For example, effectors released
by insects could be trapped within the extracellular Si matrix, precluding them from deregulating the plant
defense response, or from recognizing the plant as a suitable host (Coskun et al., 2019).
While researchers continue to debate the biological roles of Si and uncover the mechanisms behind them
(Frew et al., 2018; Coskun et al., 2019), we conclude that, at a minimum, Si mitigates the negative impacts
of various stressors, such as herbivory, which then enable plants to improve growth potential (Johnson et
al., 2019).
**4 Effects of herbivory on plant silicon**
Of the 119 reviewed studies, the majority of Si-herbivory publications have focused on insects that damage
shoots (68), while other animals including mammals (35), and insects from other feeding guilds (14), are
less frequently represented (Fig. 1). While many studies in the literature review investigate the effects of



Si on herbivory (114), few studies focus on the effects of herbivory on Si (5). Though few, these studies
show that herbivores can also induce Si uptake and accumulation by plants in response to herbivore attack
(Massey et al., 2007; Hartley and Gabriel, 2016). Although how much Si uptake is induced may depend on
plant species, herbivore type, and environmental conditions, the degree of induction can be positively
correlated with herbivory duration or frequency (Soininen et al., 2013). In one study, the Si content of two
species of grasses that experienced repeated damage by voles and locusts was 2-4 times more than
individuals of the same species that experienced only one damage event (Massey et al., 2007). Some grass
species have been shown to have as much as twice the Si contents in more heavily grazed localities
(Brizuela et al., 1986). However, one study investigating grazing effects on Si yielded different results. In a
saltmarsh, Si export rates at sheep-grazed sites were actually lower than at ungrazed sites (Müller et al.,
2013). Variable responses between different grasslands underscore the need for wider sampling to study
Si-herbivory dynamics in different ecosystems with their unique characteristics.
**4.1 Potential effects of herbivory on terrestrial silicon cycling**
While multiple reviews synthesize terrestrial Si biogeochemical cycling (*e.g.*, Conley, 2002; Struyf and
Conley, 2012) and effects of Si on herbivory (*e.g.*, Debona et al., 2017; Frew et al., 2018), few studies have
explored the potential effects of herbivory on Si cycling. Variation in Si accumulation and deposition
associated with herbivory, Si availability, and environmental variables could have important implications
for Si cycling (Cooke and DeGabriel, 2016 and references therein). Herbivores can distribute large
quantities of resources across the landscape, having important effects on nutrient cycling and ecosystem
productivity (Metcalfe et al., 2014; Bakker et al., 2016). Schoelynck et al. (2019), for example, found that
hippos contribute 32% to the biogenic Si flux and more than 76% to the total Si flux in a savannah-river
system. We estimate that Si fluxes via the herbivory pathway can meet or exceed other major sources of
Si, although flux information on some major habitat types is missing (Table 1). Herbivores may also
influence Si pathways by making more labile forms of Si available. For example, Vandevenne et al. (2013),



found that grazing by cattle can increase reactivity and dissolvability of biogenic Si after digestion, leading
to higher Si turnover rates and mobilization potential (2 versus 20 kg Si ha$^{-1}$ y$^{-1}$). It is, however, unclear
whether the more mobilized Si is then absorbed by vegetation, taken up by microbes, or exported from
the system (Fig. 3). Future research could follow the fate of more mobilized Si derived from herbivores,
which may depend on local biotic and abiotic conditions such as soil properties or plant/microbial
composition. In a wetland study, litter decomposition by heterotrophic microbes was significantly
influenced by the Si availability during plant growth, whereby litter decomposition rates were positively
correlated with higher Si content (Schaller et al., 2013). In other words, potentially greater export and
plant uptake of biogenic Si due to herbivory may impact decomposition and nutrient cycling in some
systems. Grasslands demonstrate a high capacity to store biogenic Si as well as transport Si from relatively
inert mineral silicate soil layers to biologically active soil layers (Blecker et al., 2006). If grazing can change
the distribution and reactivity of Si in grasslands, herbivory may alter Si turnover and export at rates
important to estimate for Si cycling in other agricultural and natural systems as well. Understanding the
role of herbivores in mobilizing Si may have important implications for land management.
Researchers have found that even small shifts in terrestrial biogenic Si reactivity could alter the balance
between Si storage and export from ecosystems (Struyf and Conley, 2012). Herbivory may increase
dissolved Si mobility (Vandevenne et al., 2013), and, if absorbed by plants, thereby decrease C uptake
(Cooke and Leishman, 2012), increase N use efficiency, and increase P uptake of plants (Neu et al., 2017;
also see Fig. 3). Grassland herbivores may also preferentially consume plants with less Si (Massey and
Hartley, 2006), potentially influencing ecosystem Si cycling and plant community dynamics (Garbuzov et
al., 2011). Combined with Si-mediated changes in plant nutrient stoichiometry and efficiency (Neu et al.,
2017), these findings highlight the need for more field research on how herbivory-Si interactions impact
community- and ecosystem-level processes.
**5 Conclusions**



We have begun to understand the magnitude of impact of plant Si on herbivore populations and the
potential impact of herbivores on Si fluxes, which may have important agricultural and land management
implications. In an effort to improve land management decisions and projections of biogeochemical cycling
to future climate and land-use changes, however, we need to expand our understanding of Si-herbivore
dynamics. Long-standing research makes it clear that Si plays an important role in both biogeochemistry
cycling and herbivory but major knowledge gaps remain. Based upon our review we highlight the following
future research priorities:
• Fate of more mobilized Si as a result of herbivory in agricultural settings.
• How Si-herbivory dynamics operate beyond crops in controlled or agricultural settings.
• Si-herbivore dynamics in herbivore feeding guilds other than shoot feeders.
• Impact of herbivory on biogeochemical cycling in natural settings, which remain understudied but
where evidence indicates that Si can cycle at a high rate and herbivory is an important ecosystem
process.
• Field-based studies on Si-herbivory dynamics along key environmental gradients by different
herbivore feeding guilds.
Given the demonstrated importance of Si and herbivores, and the relative paucity of information on their
interaction particularly in natural, non-graminoid dominated systems, we believe this information is
critical to generating more accurate model representations of animal-plant-soil feedbacks, and their
impacts upon ecosystem processes in different terrestrial systems.
**Data Availability**
The list of publications reviewed and classified for this article can be found at doi:
10.6084/m9.figshare.12026997.
**Authors' Contributions**





BH and DM conceived the ideas and designed methodology; BH collected and synthesized the data; With
input from DM, BH led the writing of the manuscript. Both authors contributed critically to the drafts and
gave final approval for publication.
**Competing Interests**
The authors declare that they have no conflict of interest.
**Acknowledgements**
We thank Daniel Conley for helpful comments on an early version of the manuscript. This manuscript was
written with financial support from the European Research Council (ECOHERB, grant no. 682707).

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





**Tables and Figures**

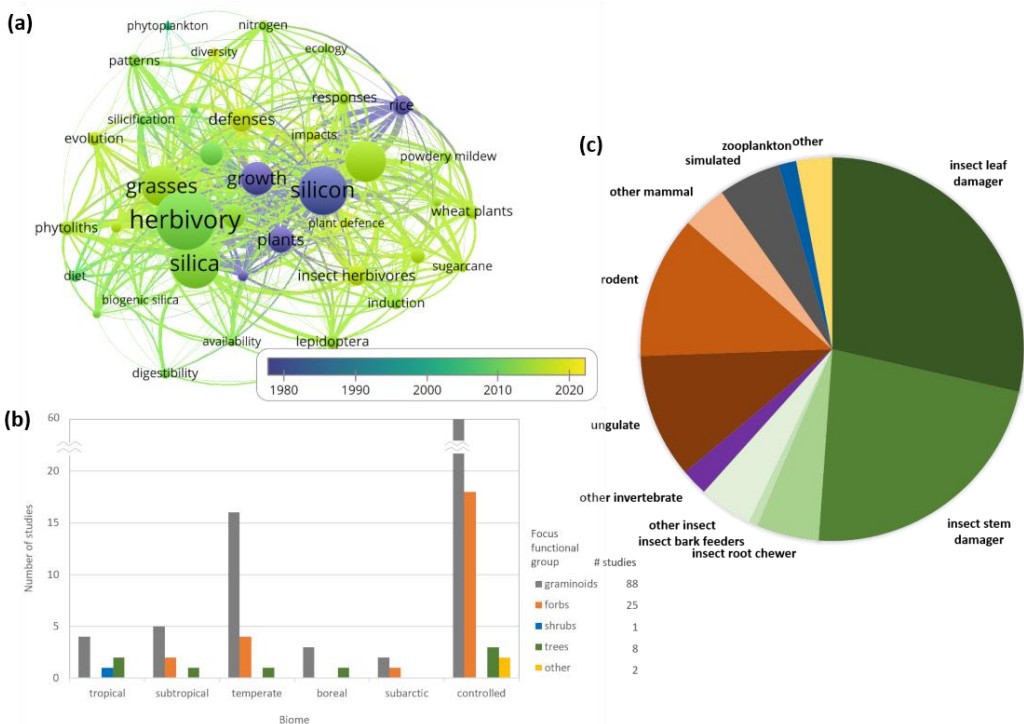


**Figure 1** Summary of literature review. **(a)** Network map of keywords from publications between 1900-
2020 in the Web of Science core collection database generated using search terms "silic*" and "herbivor*"
and not "in silico" (314 results). Bubble and word size is scaled to the total occurrence of the keywords in
all publications (only > 10 occurrences displayed), more proximate bubbles/keywords co-occur more often
in the publications surveyed. The year indicates the average publication year of the documents in which a
keyword occurs. The map was created with VOSviewer software (van Eck and Waltman 2018). **(b)** Number
of publications investigating each plant functional type and biome using the three search terms. We
filtered the 314 results until only those publications that directly studied Si and herbivory remained (119
publications). **(c)** Percentage distribution of herbivore types for all 119 studies that directly investigated



silicon and herbivory. The list of data sources and classifications for Fig. 1 are archived at doi:
10.6084/m9.figshare.12026997.





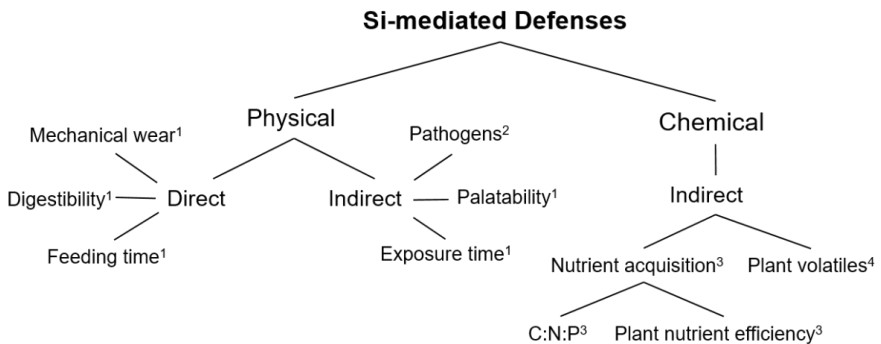


**Figure 2** Direct and indirect physical and chemical plant Si-mediated defenses to herbivory [1]Massey et al.,
2009 and references therein; [2]Lev-Yadun and Halpern, 2019; [3]Neu et al., 2017; [4]Liu et al., 2017.



**Table 1** Estimated Si fluxes by atmosphere, mineral weathering and herbivory by major terrestrial habitat
type. ANPP is aboveground net primary productivity.  [1]Cebrian 2004; [2]Hodson et al., 2005; [3]as calculated
by Carey and Fulweiler 2012; [4]Street-Parrott and Barker 2008 and references therein; [5]modeled by
Ribeiro et al., 2019; [6]Ariyanto et al., 2019.

| Habitat type[1] | Herbivory[1] (% ANPP) | ANPP[1] (g C m$^{-2}$ y$^{-1}$) | % Si by dry weight[2,3] | Si:C[3] | Herbivory (g Si m$^{-2}$ y$^{-1}$) | Atmospheric[4] (g Si m$^{-2}$ y$^{-1}$) | Weathering[4] (g Si m$^{-2}$ y$^{-1}$) |
|---|---|---|---|---|---|---|---|
| Tundra shrublands and grasslands | 1.8 ± 0.6 | 95.3 ± 31.0 | 1.07 | 0.02 | 0.04 | - | - |
| Temperate/tropical shrublands and forests | 7.9 ± 1.5 | 334.6 ± 24.9 | 0.26 | 0.01 | 0.15 | 0.004 - 0.2 | 0.3-3.2 |
| Temperate/tropical grasslands | 34.1 ± 5.3 | 239.4 ± 34.0 | 1.33 | 0.03 | 2.31 | 0.2 | 0.4-2.5 |
| Freshwater and marine marshes | 16.8 ± 5.0 | 958.9 ± 179.0 | 0.62 | 0.01 | 2.13 | - | - |
| Mangroves | 3.5 ± 0.5 | 500[5] | 0.43[6] | 0.01 | 1.60 | - | - |






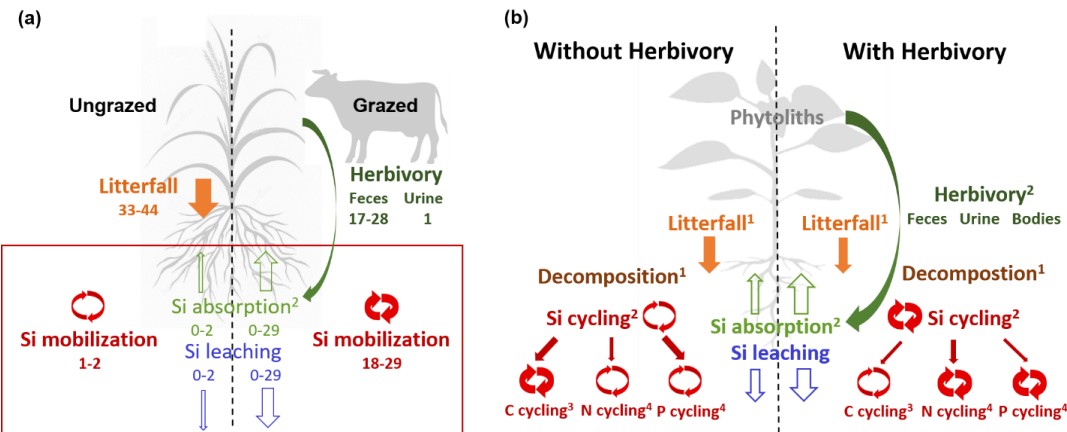

**Figure 3 (a)** Schematic of Si fluxes (kg ha$^{-1}$ yr$^{-1}$) in hypothetical ungrazed and grazed grassland, dominated
by cows, adapted from Vandevenne et al., 2013. Annual net primary production is assumed to be the same
for both systems (11,000 kg ha$^{-1}$ yr$^{-1}$) and all biomass is converted to litter or consumed by grazers in the
grazed system. A biomass conversion ratio of 1 : 4 for grass versus feces is suggested for all herbivores.
Mobilization percentages for faeces are based on ranges (minimum - maximum values) obtained in a
dissolution experiment by Vandevenne et al., 2013 after 24 h in rain water; for urine, a conservative
estimation of 3% mobilization was used. **(b)** Schematic of potential nutrient cycle differences in the plant-
soil system with and without herbivory. Size of arrows denotes the size of the flux relative to the
alternative scenario. [1]Schaller et al., 2016 [2]Brizuela et al., 1986; [3]Street-Perrott and Barker 2008; [4]Neu et
al., 2017; Schaller et al., 2017.