# Peer review of "Reviews and Syntheses: Impacts of plant silica - herbivore interactions on terrestrial biogeochemical cycling Authors"

_Biogeosciences, 2020_

## Referee Comment (RC1) · Anonymous Referee #1 · 16 Nov 2020

An interesting and timely paper on a neat concept - bringing together information about plant Si, herbivory and terrestrial biogeochemical fluxes. Table 1 and Figure 3 are of most interest - and they could be made more of in the text, to better convey and highlight the syntheses the authors have done (see below for suggestion on how).

After the introduction, Section 2 is about Silicon in terrestrial systems, and 3 Effects of silicon on herbivory, 4 Effects of herbivory on plant silicon, 4.1 Potential effects of herbivory on terrestrial silicon cycling and 5 Conclusions. The information providing sections 2, 3 and 4 make up a lot of the manuscript with the syntheses (4.1 and 5) a smaller component. One way to keep the focus on the synthesis, would be to re-

[Figure]

move recommendations that are not clearly/directly associated with the synthesis (ie. in Sections 2 and 3). For example, Line 144 "Therefore, we need more field-based information about how Si content varies along large-scale environmental gradients to improve global biogeochemistry models." – this might well be true, but it is a call to action on a topic that isn't related to the paper, as it appears in Section 2 (that doesn't mention herbivory at all). See also lines 158 and worth checking elsewhere.

Also, elevating section 4.1 to 5 (and renumbering the conclusion) would also help highlight the significance of this section. In 4.1 to make more of the key data (Fig 3, Table 1), they could be referred to in a more positive way. For example, for L220, could be changed from "We estimate that Si fluxes via the herbivory pathway can meet or exceed other major sources of Si, although flux information on some major habitat types is missing (Table 1)." to "Bringing together published Si flux data with estimates of herbivory for the first time, we estimate that Si fluxes via the herbivory pathway could meet or exceed other major sources of Si (Table 1), although flux information on some major habitat types is missing." Similarly there is much more to Figure 3 there than just "It is, however, unclear whether the more mobilized Si is then absorbed by vegetation, taken up by microbes, or exported from the system (Fig. 3)." which is how it is first referred to.

The conclusion too, could better highlight what the author's have done. For example, L247: "We have begun to understand the magnitude of impact of plant Si on herbivore populations and the potential impact of herbivores on Si fluxes" would be stronger as "Our analysis has shown the magnitude of impact of plant Si on herbivore populations and the potential impact of herbivores on Si fluxes"

Minor comments: L90: soil -> the soil. Also. there is a lot in this sentence with both Si recycling rates and introducing occluded C in phytoliths. Suggest breaking into two sentences. L94-96: can it also increase fluxes to rivers? For example, when leaves from deciduous forests fall, are there spike in Si fluxes to water ways? Figure 3b. Check arrow size between Si cycling and P cycling on the left hand side of the figure.

---

## Referee Comment (RC2) · Jonas Schoelynck (Referee) · 8 Dec 2020

I was very pleased to read the paper. The topic is very timely and interesting and the text is pleasant to read. I also think that the different subtopics/paragraphs are well-balanced and cite most of the relevant publications. The meta-analysis of the literature was good for what it is, but not even necessary. The main message is strong enough to publish on its own, but I guess you need it to prove your claim. I whish I came up with the idea of this review/opinion paper :o)

I'd love to see this paper published as is.

Cheers

Jonas

---

## Author Comment (AC1) · 18 Dec 2020

(Comment) An interesting and timely paper on a neat concept - bringing together information about plant Si, herbivory and terrestrial biogeochemical fluxes. Table 1 and Figure 3 are of most interest - and they could be made more of in the text, to better convey and highlight the syntheses the authors have done (see below for suggestion on how). (Response) We thank the referee for their overall positive assessment of the manuscript and suggestions. We have now revised the manuscript according to the referee's comments (see specific responses below), which we believe has further improved the manuscript. Line numbers refer to a revised version of the manuscript that

we hope to be permitted to upload.

(Comment) After the introduction, Section 2 is about Silicon in terrestrial systems, and 3 Effects of silicon on herbivory, 4 Effects of herbivory on plant silicon, 4.1 Potential effects of herbivory on terrestrial silicon cycling and 5 Conclusions. The information providing sections 2, 3 and 4 make up a lot of the manuscript with the syntheses (4.1 and 5) a smaller component. One way to keep the focus on the synthesis, would be to remove recommendations that are not clearly/directly associated with the synthesis (ie. in Sections 2 and 3). For example, Line 144 "Therefore, we need more field-based information about how Si content varies along large-scale environmental gradients to improve global biogeochemistry models." – this might well be true, but it is a call to action on a topic that isn't related to the paper, as it appears in Section 2 (that doesn't mention herbivory at all). See also lines 158 and worth checking elsewhere. (Response) We thank the referee for the recommendation and have made the suggested change by eliminating the highlighted sentences.

(Comment) Also, elevating section 4.1 to 5 (and renumbering the conclusion) would also help highlight the significance of this section. In 4.1 to make more of the key data (Fig 3, Table 1), they could be referred to in a more positive way. For example, for L220, could be changed from "We estimate that Si fluxes via the herbivory pathway can meet or exceed other major sources of Si, although flux information on some major habitat types is missing (Table 1)." to "Bringing together published Si flux data with estimates of herbivory for the first time, we estimate that Si fluxes via the herbivory pathway could meet or exceed other major sources of Si (Table 1), although flux information on some major habitat types is missing." Similarly there is much more to Figure 3 there than just "It is, however, unclear whether the more mobilized Si is then absorbed by vegetation, taken up by microbes, or exported from the system (Fig. 3)." which is how it is first referred to. (Response) We have promoted section 4.1 to 5 (L212) and renumbered the conclusion accordingly (L252). In addition, we have acted on Referee 1's suggestion to elaborate on and/or refer to Fig. 3 and Table 1 more positively: Bringing together

published Si flux data with estimates of herbivory for the first time, we estimate that Si fluxes via the herbivory pathway could meet or exceed other major sources of Si (Table 1), although flux information on some major habitat types is missing. Herbivores may also influence Si pathways by making more labile forms of Si available. For example, Vandevenne et al. (2013), found that grazing by cattle can increase reactivity and dissolvability of biogenic Si after digestion, leading to higher Si turnover rates and mobilization potential (2 versus 20 kg Si ha−1 y−1). Greater Si mobilization terrestrially due to herbivory can potentially affect the uptake of Si by plants as well as the movement of other linked nutrients indirectly (Fig. 3). It is, however, currently unclear whether the more mobilized Si is absorbed by vegetation, taken up by microbes, or exported from the system (L221-231).

(Comment) The conclusion too, could better highlight what the author's have done. For example, L247: "We have begun to understand the magnitude of impact of plant Si on herbivore populations and the potential impact of herbivores on Si fluxes" would be stronger as "Our analysis has shown the magnitude of impact of plant Si on herbivore populations and the potential impact of herbivores on Si fluxes" (Response) We thank the reviewer the recommendation and have made the suggested change to the introductory sentence of the conclusion (L253-356).

(Comment) Minor comments: L90: soil -> the soil. Also. there is a lot in this sentence with both Si recycling rates and introducing occluded C in phytoliths. Suggest breaking into two sentences. (Response) We have split the sentence in the following way: "Si then returns to the soil when plant material decomposes either as dissolved Si, a quickly-available source of Si for terrestrial plants, or remain as as phytoliths. Carbon incorporated by phytoliths may accumulate in soils and sediments for hundreds to thousands of years." (L90-93)

(Comment) L94-96: can it also increase fluxes to rivers? For example, when leaves from deciduous forests fall, are there spike in Si fluxes to water ways? Figure 3b. (Response) Yes, aquatic systems can also be affected by the potential increase in Si

flux. We have edited the sentence to the following: "Plant-accumulated Si has been shown to reduce the magnitude of Si released from terrestrial to aquatic ecosystems, thereby having direct implications on Si availability in rivers and coastal waters, which could the influence diatom blooms and C uptake rates (Coney et al., 2008, Carey and Fulweiler, 2012; see also Fig. 3b)." (L94-97)

(Comment) Check arrow size between Si cycling and P cycling on the left hand side of the figure. (Response) Thanks to Referee 1 for pointing out this detail. We have decreased the arrow size between Si cycling and P cycling on the left-hand side of the figure (L520).
* * *
[Figure]

[Figure]

**Fig. 1.**

---

## Author Comment (AC2) · 18 Dec 2020

Bernice C. Hwang and Daniel B. Metcalfe

bernice.hwang@nateko.lu.se

(Comment) I was very pleased to read the paper. The topic is very timely and interesting and the text is pleasant to read. I also think that the different subtopics/paragraphs are well-balanced and cite most of the relevant publications. The meta-analysis of the literature was good for what it is, but not even necessary. The main message is strong enough to publish on its own, but I guess you need it to prove your claim. I whish I came up with the idea of this review/opinion paper :o) I'd love to see this paper published as is.

(Response) We thank the reviewer for his positive assessment of the manuscript. We

hope that the manuscript will encourage more research in this overlooked topic.